# Anti-Inflammatory Potential of the Anti-Diabetic Drug Metformin in the Prevention of Inflammatory Complications and Infectious Diseases Including COVID-19: A Narrative Review

**DOI:** 10.3390/ijms25105190

**Published:** 2024-05-10

**Authors:** Trevor J. Plowman, Hannah Christensen, Myia Aiges, Emely Fernandez, Mujtaba H. Shah, Kota V. Ramana

**Affiliations:** Department of Biomedical Sciences, Noorda College of Osteopathic Medicine, Provo, UT 84606, USA

**Keywords:** metformin, inflammation, COVID-19, diabetes, oxidative stress

## Abstract

Metformin, a widely used first-line anti-diabetic therapy for the treatment of type-2 diabetes, has been shown to lower hyperglycemia levels in the blood by enhancing insulin actions. For several decades this drug has been used globally to successfully control hyperglycemia. Lactic acidosis has been shown to be a major adverse effect of metformin in some type-2 diabetic patients, but several studies suggest that it is a typically well-tolerated and safe drug in most patients. Further, recent studies also indicate its potential to reduce the symptoms associated with various inflammatory complications and infectious diseases including coronavirus disease 2019 (COVID-19). These studies suggest that besides diabetes, metformin could be used as an adjuvant drug to control inflammatory and infectious diseases. In this article, we discuss the current understanding of the role of the anti-diabetic drug metformin in the prevention of various inflammatory complications and infectious diseases in both diabetics and non-diabetics.

## 1. Introduction

Metformin (1,1-dimethyl biguanide hydrochloride) has been one of the most prescribed anti-diabetic drugs to control type-2 diabetes worldwide for several decades [1]. It is a synthetic derivative of galegine and guanidine, which is naturally found in the plant *Galega officinalis* (French Lilac [2]). Although metformin has been shown to decrease glucose absorption, reduce gluconeogenesis, and increase glucose uptake, the mechanisms through which this compound lowers plasma glucose levels are still unclear [3,4,5]. However, few studies indicate that metformin’s actions are mediated through AMP-activated protein kinase (AMPK)-dependent and -independent pathways [6,7]. AMPK is a central regulator of several intracellular metabolic pathways including lipid and carbohydrate metabolism, cellular respiration, and energy production. AMPK is also involved in insulin sensitivity and signaling in the liver, reducing insulin-mediated hepatic gluconeogenesis [8]. Further, AMPK is also involved in regulating mitochondrial respiratory chain complex-1, and metformin has been shown to inhibit electron transport chain complex-1 by unknown mechanisms [9]. Inhibition of mitochondrial complex-1 leads to a decrease in adenosine triphosphate (ATP) levels and increased adenosine monophosphate (AMP) levels, which could activate AMPK [9].

Further, metformin also prevents the activation of fructose-1,6-bisphosphate, thereby inhibiting hepatic gluconeogenesis. Increased AMPK has also been shown to increase the expression of small heterodimer partners that inhibit glucose-6-phosphatase and phosphoenolpyruvate carboxykinase [10]. Further, Miller et al. 2013 [11] reported that metformin blocks liver glucose levels independent of AMPK by increasing cyclic AMP (cAMP) levels; moreover, metformin could decrease adenylate cyclase and glucagon-induced cAMP levels. Similarly, metformin disrupts cellular redox balance by inhibiting glycerol-3-phosphate dehydrogenase, which is involved in the oxidation reaction of glycerol-3-phosphate to dihydroxyacetone where FADH2 is generated [12]. This reaction facilitates the transfer of electrons from FADH2 to coenzyme Q of the mitochondrial electron transport chain during carbohydrate metabolism.

Metformin has also been shown to increase the secretion of glucagon-like peptide-1 (GLP1) in the gut, which is responsible for its glucose-reducing actions. A recent study by Ma et al. [6] showed that low levels of metformin inhibit the lysosomal proton pump v-ATPase via presenilin enhancer 2 (PEN2), a subunit of gamma-secretase, in an AMP-independent AMPK activation. The study indicated that metformin targets PEN2, thereby interrupting the PEN2-ATP6AP1 axis, which disrupts the glucose-sensing pathway to activate AMPK. Another study by Kjobsted et al. [13] indicated that in a lean and diet-induced obese mouse model, metformin increased intestinal glucose clearance without involving skeletal muscle AMPK activation. This study also indicated that chronic metformin increases skeletal muscle AMPK activity in type-2 diabetic patients, which is not associated with increased peripheral insulin sensitivity. Thus, these studies indicate that metformin exerts glucose-lowering effects via AMPK-dependent and -independent mechanisms, depending on the concentrations of metformin used.

Metformin has been shown to prevent insulin resistance and associated metabolic syndrome. Insulin resistance, a condition characterized by a weakened response to insulin in the cells, plays a crucial role in the development of metabolic syndrome and diabetes [14]. Insulin resistance is associated with the increased production of pro-inflammatory cytokines [15]. Metformin has been shown to control insulin resistance by activating AMPK, thereby increasing insulin sensitivity and peripheral glucose uptake [16]. Further, metformin has been shown to control diabetes by promoting insulin receptor expression and insulin signaling by activating tyrosine kinase [17]. Several studies also suggest that metformin prevents hyperglycemia by increasing the expression of glucagon-like peptide 1 (GLP1) via peroxisome proliferator-activated receptor (PPAR)-α [18,19]. Thus, multiple studies have clearly demonstrated that metformin prevents insulin resistance and hyperglycemia, mostly through inhibition of mitochondrial respiratory complex 1, activation of AMPK, and enhancement of glucose uptake (Figure 1).

Due to its basic hydrophilic nature, metformin exhibits distinctive pharmacokinetics. Its oral bioavailability ranges from 40–60%, with absorption primarily occurring in the small intestine and peak plasma concentrations reached within 2 to 3 h. Although its intestinal absorption is minimal via simple diffusion, several transport proteins such as organic cation transporters (OCT1 and 3) actively facilitate metformin transportation through the intestinal plasma membrane. In contrast to several xenobiotic drugs, metformin neither binds to plasma proteins nor undergoes hepatic metabolism. It is predominantly eliminated through the kidneys, with renal clearance exceeding the glomerular filtration rate (GFR) via OCT transporters. Renal dysfunction leading to decreased metformin clearance could potentially result in lactic acidosis. For additional reading, please refer to the recent article by Froldi, 2024 [20]. Although lactic acidosis is one of the side effects of metformin in some diabetic patients, most patients tolerate metformin use without significant lactic acidosis. Further, the benefits of metformin have outweighed this minor side effect, contributing to its continued prescription for the control of type-2 diabetes to date. Indeed, metformin has been continuously prescribed to control type-2 diabetes for several decades [1,2,3,4]. This narrative review article discusses the therapeutic benefits of this drug in controlling various complications besides diabetes.

## 2. Methodology

Studies from the last decade or so have indicated the significant therapeutic use of this drug in controlling various diseases other than diabetes, including inflammatory complications, cancer, and infectious diseases, in both diabetic and non-diabetic conditions. The objective of this article is to help readers understand the current role of metformin in regulating inflammatory and infectious diseases. We carried out a PubMed search to find articles published in the last 10 years. We used keywords such as metformin, diabetes, COPD, asthma, sepsis, cardiovascular complications, infections, HIV, and COVID-19. In addition, we included full-text research articles, meta-analysis studies, systematic reviews, and clinical and pre-clinical studies in our search. We excluded publications that show the combined effect of metformin with other known anti-diabetic drugs.

## 3. Metformin in Controlling Respiratory Complications

Chronic respiratory diseases such as asthma and chronic obstructive pulmonary disease (COPD) have become some of the most significant burdens on healthcare systems worldwide, representing a growing cause of mortality worldwide. COPD is the fourth leading cause of death, according to the World Health Organization [21]. In addition to genetic predispositions and lifestyle factors, infectious diseases also play an essential role in the pathophysiology of both asthma and COPD. Viral infections, particularly with Rhinovirus and Respiratory Syncytial Virus (RSV) during childhood, have been linked with the development of asthma and the exacerbation of the disease later in life [21,22]. Moreover, repeated respiratory infections during childhood, along with previous tuberculosis infections, have been shown to be significant non-smoking risk factors in the development of COPD and the worsening of its symptoms [23]. Recently, the high prevalence of coronavirus disease 2019 (COVID-19) has also become relevant to developing inflammatory respiratory complications such as pneumonitis.

Current treatments for COPD utilize glucocorticoids and bronchodilators to provide symptomatic relief but fall short of effectively stopping the progression of the disease, which presents a clear limit to their clinical efficacy [24]. Treatments for asthma are primarily interventional and focus on managing exacerbations using inhaled corticosteroids. Currently, the prevention of symptoms remains a largely unmet need in asthma management [25]. While the ongoing management of COVID-19 remains a highly debated topic, symptom prevention centers around effective vaccination of the individual against SARS-CoV-2. Symptoms vary, but infected patients are treated most often with oxygen therapy, antivirals, antibiotics, and corticosteroids. Notably, macrolides such as azithromycin can effectively prevent pulmonary infections in patients with viral pneumonitis and have a significant anti-inflammatory effect on the airways [26].

Recent studies indicate that metformin shows promise as a treatment option for various respiratory complications. Cui et al. [27] found that in a cigarette smoke-induced COPD model in mice, metformin treatment increased the expression of Wnt3α, β-catenin, and Nrf2 and decreased the pro-inflammatory cytokines such as IL-6 and IL-8. Further, a recent study by Polverino et al. [28] extensively investigated the beneficial effects of metformin in a chronic cigarette smoke-induced mouse model of airway inflammation. This study indicates that oral administration of metformin prevents chronic cigarette smoke-induced airspace enlargement, lung inflammation, small airway remodeling, oxidative stress, mitochondrial dysfunction, and lung, kidney, and muscle injury. Furthermore, Gu et al. [29] have shown that metformin prevents non-allergic airway hyperresponsiveness in a mouse model of diet-induced obesity. Similarly, metformin has been shown to prevent eosinophil infiltration, expression of AMPK, eotaxin, TNF-α, and formation of (nitric oxides) NOx in a high fat-diet-induced and ovalbumin-challenged mouse model of asthma [30]. Metformin has also been shown to reduce allergic airway inflammation by increasing the Treg cells in an obese asthma mouse model [31]. Metformin has also been shown to prevent airway hyperreactivity in rat models of obesity-related asthma and COPD in cigarette smoke-induced rats [32]. These studies indicate that metformin could prevent asthma and COPD in animal models in an AMPK-dependent and independent manner (Figure 2). While there are few clinical studies demonstrating the significance of metformin in controlling respiratory complications, the preclinical evidence is promising. 

The National Health Insurance Research Database in Taiwan conducted a retrospective cohort study that examined the efficacy of treating asthma with metformin [33]. The study found that patients taking metformin were less likely to be hospitalized with asthma-related symptoms and less likely to experience worsening asthma symptoms when compared to the group of non-metformin users. The anti-inflammatory effect that metformin had on the airways was attributed to the inhibition of eosinophilic inflammation due to increased activation of AMP kinase. The same mechanism could also inhibit the inflammatory process in other diseases like cystic fibrosis, colitis, and lipopolysaccharide-induced lung inflammation resulting from bacterial infections [33].

Similarly, a claims-based cohort study by Wu et al. [34] indicated that in a cohort of 23,920 subjects with asthma and diabetes, metformin lowers asthma exacerbation and hospitalizations. A multicentered, randomized, double-blind, and placebo-controlled clinical study examined metformin therapy in COPD in non-diabetic patients. This study found no significant difference between the metformin and placebo-treated group in controlling the COPD assessment scores [35]. Another retrospective analysis of a multi-center observation cohort COPD study indicated that metformin reduces respiratory exacerbations in patients with physician-diagnosed asthma. Further, this study suggested that metformin increases the quality of life in physician-diagnosed asthma patients before the age of 40 [36].

Further, Xian et al. [37] tested the efficacy of metformin treatment in acute respiratory distress syndrome, a dangerous inflammatory condition caused by respiratory bacteria and viruses, which is common in severe cases of COVID-19 infection. Mice were injected with 50 mg/Kg of metformin two days before inoculation with SARS-CoV-2, and lung tissue was then collected for analysis six days post-infection. The study found that metformin treatment inhibited the infiltration of inflammatory cells into the bronchioles, vascular bed, and lung parenchyma while also preventing alveolar wall thickening, effectively attenuating SARS-CoV-2-induced pulmonary inflammation, and its related symptoms [37]. Although a clear mechanism is not entirely understood, metformin has been found to non-specifically inhibit NLR family pyrin domain containing 3 (NLRP3) inflammasome activation and subsequent secretion of IL-1β by targeting the electron transport chain complex 1 and blocking the generation of oxidized mitochondrial DNA. Additionally, it inhibits the inflammasome-independent transcription of IL-6 [38]. The cytokines IL-1β and IL-6 are significant promoters of inflammation and mediate innate and adaptive immunity; inhibition of their production attenuates these responses and prevents the complications of acute respiratory distress syndrome [39].

## 4. Metformin in Controlling Cardiovascular Complications

Cardiovascular complications, such as atherosclerosis and cardiomyopathy, are primarily influenced by lifestyle, family history, and environment. Further, specific populations are at increased risk of developing these complications due to their dietary intake and the prevalence of diabetes. Hyperglycemia is one of the most common risk factors for developing a cardiovascular illness, and cardiovascular diseases rank among the major causes of morbidity and mortality in patients with type-1 and type-2 diabetes [40,41]. Indeed, a study by Dandamudi et al. [42] found that among diabetic patients, 16.9% met the criteria for developing diabetic cardiomyopathy. They suggest that diabetes is associated with a 1.9-fold increase in the risk of developing left ventricular dysfunction, a 1.7-fold increase in the risk of developing diastolic dysfunction, and a 2.2-fold rise in the risk of developing systolic dysfunction [42]. Atherosclerosis, characterized by the buildup of plaque within blood vessels, can lead to severe injuries to subcutaneous tissues and organs, depending on the location of the plaque. Hyperglycemia can increase the risk of developing plaques. A population-based study by Resnick et al. [43] indicated that the first myocardial infarction rate for patients with diabetes was ~20% when compared to non-diabetics (3.5%). In addition, patients with diabetes are more at risk for stroke, peripheral arterial diseases, and congestive heart failure.

Controlling diabetes, engaging in physical activity, maintaining low cholesterol levels, and adopting a healthy lifestyle are preventive strategies for reducing the occurrence of cardiovascular complications. Metformin could act as a cardioprotective agent through its potent antioxidative and anti-inflammatory functions. Specifically, in diabetic patients, metformin has been shown to prevent the morbidity and mortality associated with cardiovascular complications [44,45].

Vascular endothelial cells play a critical role in maintaining cardiovascular system homeostasis [46]. These cells provide a physical barrier between the vessel wall and the lumen. Further, the endothelium secretes several mediators, such as inflammatory cytokines, adhesion molecules, nitric oxide, and prostaglandins, that regulate platelet aggregation, coagulation, fibrinolysis, and vascular tone. Dysfunction of the endothelium, leading to loss of its normal physiological function, has been implicated in the pathology of different cardiovascular diseases, such as hypertension, coronary artery disease (CAD), chronic heart failure, and peripheral artery disease [46,47]. Various studies have indicated that metformin prevents endothelial dysfunction both in vitro and in vivo. Han et al. (2018) showed that metformin prevents high glucose-induced endothelial dysfunction and death via AMPK, CREB, and BDNF-dependent mechanisms [48]. Similarly, Chen et al. [49] have demonstrated that metformin prevents bevacizumab-induced endothelial dysfunction by activating the PI3K/AKT/FOXO/PPAR-γ pathway in human umbilical vascular endothelial cells (HUVECs). Further, Detaille et al. [50] have shown that metformin prevents endothelial cell death by regulating the mitochondrial permeability transition pore. Metformin also prevents hyperglycemia-induced endothelial dysfunction by reducing the formation of reactive oxygen species. It improves vascular function by expressing orphan nuclear receptor NR4A1/Nur77 in a streptozotocin (STZ)-induced diabetic mouse model [51]. Similarly, Chellian et al. [52] have also shown that metformin prevents endothelial dysfunction in STZ/nicotinamide-induced diabetic rat aortas.

Several studies have shown the anti-atherosclerotic effects of metformin in preclinical models (Figure 3). In addition to its anti-hyperglycemic properties, metformin prevents adipose tissue lipolysis, circulation of free fatty acids, formation of reactive oxygen species (ROS), formation of advanced glycation end products (AGEs), and increased insulin sensitivity and signaling. These properties of metformin help control and develop endothelial dysfunction and cardiovascular complications. For example, metformin has been shown to exert cardioprotective function by reducing free fatty acid levels in diabetic rats [53]. Similarly, Majithiya et al. [54] have shown that metformin prevents STZ-induced increases in blood pressure and endothelial dysfunction in diabetic rats. Metformin also prevents atherosclerotic lesions in the STZ-induced diabetic ApoE null mice model by increasing the AMPK1 levels and reducing the free radical formation and expression of dynamin-related protein [55].

Several clinical studies have also examined the efficacy of metformin in preventing cardiovascular complications in both diabetics and non-diabetics. A multicentered, randomized, double-blind, placebo-controlled study indicated that three years of metformin treatment significantly reduced cardiovascular complications in diabetic patients when compared to glipizide [56].

Similar results were observed in additional clinical studies involving patients with type-2 diabetes [56,57]. Matsumoto et al. [58] investigated the anti-atherogenic actions of metformin in type-2 diabetic patients. In this open prospective study, they found that the progression of carotid intima–media thickness was significantly reduced in patients treated with metformin.

## 5. Metformin in Controlling Ocular Inflammatory Complications

Ocular inflammatory complications, such as uveitis and retinitis, are the leading causes of vision impairment in both non-diabetic and diabetic subjects [59,60]. Increased levels of inflammatory cytokines, chemokines, and growth factors in aqueous and vitreous humor could cause significant damage to adjacent tissues and lead to visual impairment. In some pre-clinical studies of retinal diseases involving animals or cell lines, metformin has been shown to exert a therapeutic benefit in treating various ocular inflammatory conditions. These studies indicate that metformin has a protective effect on retinal diseases through a broad mechanism of action, including anti-inflammatory, antioxidative, and antiangiogenic pathways. Regarding the potential anti-inflammatory effect of metformin, it seems to be facilitated through the inhibition of NF-κB signaling and the reduction in pro-inflammatory cytokines in ocular tissues. Additionally, metformin also inhibits leukocyte activation by decreasing levels of MCP-1, G-CSF, and ICAM [61,62].

The effect of metformin in controlling visual complications in diabetic retinopathy (DR) and age-related macular degeneration (AMD) patients is well established [63,64,65]. Most of these pre-clinical and clinical studies indicate that metformin prevents symptoms associated with DR and AMD. However, studies on its effects on ocular inflammatory complications are limited. Uveitis is a heterogeneous collection of diseases depicted by intraocular inflammation caused by auto-immune disorders, infections, toxins, and other factors [66]. In many forms of uveitis, the activation of redox-sensitive transcription factors, including NF-κB, leads to increased transcription of pro-inflammatory cytokines and chemokines. Kalariya et al. [67] demonstrated the anti-inflammatory effects of metformin in an endotoxin-induced uveitis rat model. It was found that treatment with metformin suppressed the LPS-induced increase in levels of cytokines and chemokines in the aqueous humor, including TNF-α, MCP-1, IL-1β, MIP-1α, IL-6, Leptin, IL-18, and GRO/KC. Immunohistochemical analysis of the epithelial cells of the ciliary body and retina revealed that metformin reduced the expression of cyclooxygenase-2 (COX-2) and phosphorylated p65 and increased the expression of phosphorylated AMPK. The authors hypothesized that by activating AMPK, metformin prevents endotoxin-induced NF-κB-dependent expression of inflammatory markers. Another study by Li et al. [68] suggests that metformin reduces the expression of pro-inflammatory markers in the retinal endothelial cells and vitreous humor in patients with proliferative diabetic retinopathy. Further, metformin also prevents retinal cell death in diabetic mouse models. Metformin has also been shown to increase AMPK activity and prevent photoreceptor and retinal endothelial cell degeneration in mice [69]. Furthermore, metformin has been shown to prevent diabetes-associated histopathological ocular deteriorations in various ocular tissues such as cornea, sclera, iris, retina, and ciliary body by preventing oxidative stress, inflammatory response, and neovascularization in a STZ-induced diabetic rat model [70]. Moreover, metformin has been shown to prevent ocular inflammation independent of its anti-diabetic actions [71].

Although the effect of metformin on ocular inflammatory complications has been investigated in animal models, very few studies are available on its significance in human ocular inflammatory complications [72]. A solitary clinical study investigating the association between metformin use and the onset of non-infectious uveitis suggests that a longer duration of metformin use was protective against the condition compared to no metformin use [73]. Further, a few meta-analysis studies indicate that prior metformin use reduces the risk of developing AMD [63,74]. In conclusion, pre-clinical studies suggest that metformin, with its potent antioxidative and anti-inflammatory effects, may be helpful in the treatment of ocular inflammatory complications such as uveitis. However, further clinical research is warranted to clarify the mechanism and efficacy in humans.

## 6. Metformin in Controlling Multiple Organ Dysfunction and Sepsis

Uncontrolled bacterial infections could lead to systemic pathology called sepsis or septicemia. Although various antibiotic treatments could control the spread of infections, the bacterial debris containing toxic endotoxins such as lipopolysaccharides (LPSs) could circulate in the bloodstream, leading to immune dysfunction and significantly impacting immune cells’ ability to control pathogen attacks. The endotoxins cause significant release of reactive oxygen free radicals that activate redox-sensitive transcription factors, leading to the expression of various pro-inflammatory cytokines, chemokines, and growth factors, which cause disseminated intravascular coagulation, immune disbalance, and life-threatening multiple organ failure [75]. Therefore, anti-inflammatory agents are warranted for treating sepsis in addition to broad-spectrum antibiotics. Regardless of recent innovations in understanding the pathophysiology of sepsis and novel treatment approaches, the mortality associated with sepsis has decreased; however, the incidence rate is still alarmingly high worldwide [76]. The mortality rate from sepsis has gone down from 37% to 30% over the last decade; however, that rate is still high [77,78]. Early infection detection and appropriate antibiotic and anti-inflammatory therapies are vital for better patient recovery and survival. Poorly controlled infections could thus lead to septic shock and multiple organ failure, leading to death. The most common biomarkers of sepsis include increased procalcitonin (early sepsis indicator), increased C-reactive protein, inflammatory cytokines (TNF-α and IL-6), monocyte chemoattractant protein-1 (MCP-1), complement pathway activation, neutrophil surface receptor (CD64), angiopoietin, MMPs, and many more to patient recovery and survival [79,80]. The most used indicator for organ failure is blood lactate levels [81]. Additionally, creatinine and bilirubin can be used to monitor kidney and liver function, respectively [82,83]. Several anti-inflammatory agents, such as corticosteroids, non-steroidal anti-inflammatory agents, and antioxidants, have been tested for their efficacy in preventing cytokine bursts during sepsis. Few studies have also indicated the significance of the anti-diabetic drug metformin in controlling inflammatory response in sepsis conditions.

Tian et al. [84] showed a possible molecular pathway in which metformin modulates the immune response to endotoxemia and sepsis. They state that lipopolysaccharide (LPS) activates endothelial cells and enhances leukocyte recruitment, leading to loss of vascular integrity. This loss in vascular integrity is a critical player in the progression of sepsis. In a mouse and cell culture model of endotoxemia, this study specifically looks at the implication of AMPK activation, performed by metformin, on histone deacetylases (HDACs) and Kruppel-like factor 2 (KLF2). HDACs are post-translational modifiers that are highly active and bound to KLF2 in sepsis. KLF2 is a transcription factor that helps maintain vascular function and decrease pro-inflammatory responses. The results of the study suggest that pre-administration with metformin increased LPS-induced HDAC5 phosphorylation, detaching it from KFL2, thereby increasing free KLF2 and subsequently decreasing VCAM1 expression.

Additionally, metformin also significantly decreased TNF-α and IL-6. Interestingly, AMPK knockout mice showed no impact on HDAC5 phosphorylation, and VCAM1 levels remained high, even with the pre-administration of metformin. Similarly, another study by Song et al. [85] observed the effects of metformin administered simultaneously with LPS injection in mice. This study investigated how metformin alters sepsis-induced liver failure and damage. They found that metformin significantly impacted immune function and cellular metabolism in addition to decreasing the degree of liver damage. In the LPS + metformin-treated group, there was a reduction in the downregulation of PGC-1α, caspase-1, IL-6, and TNF-α. Also, metformin reversed the LPS-induced increase in PDK-1 and HIF-1α, further supporting an anti-inflammatory theory. Another study by Matsiukevich et al. [86] demonstrated metformin’s age- and gender-dependent influence on the AMPK/SIRT1/PGC-1α pathway. The study induced hemorrhagic shock in age/gender-matched control, young male, young female, mature male, and mature female groups of mice without and with metformin. The results of the study found that metformin did not have a significant impact on blood lactate or glucose levels 3 h after resuscitation. Another finding showed that mature male and female groups had a decrease in IL-6 and TNF-α, among other cytokines. The young female group also had a reduction in TNF-α. Interestingly, the study did not report a distinct increase in pAMPK in the metformin groups. However, metformin did increase the nuclear translocation of PGC-1α to normal levels in all metformin groups. This finding was corroborated by increased SIRT1 in all groups treated with metformin. Thus, this study indicates that metformin, by regulating the AMPK/SIRT1/PGC-1α pathway, could reduce reactive oxygen species and inflammatory cytokines (IL-1β, IL-6, IL-8, and TNF-α) in a mouse model of sepsis (Figure 4).

Several other studies also suggest the significance of metformin in controlling sepsis in diabetic and non-diabetic preclinical models (Figure 4). Metformin has been shown to prevent neuroinflammation and injury by restoring the gut microbiota and their metabolite dysbiosis [87]. Another study suggests that long-term metformin treatment is beneficial in improving healthspan and lifespan in male mice [88]. This study suggests that metformin increases AMPK activity and antioxidant protection as well as decreases oxidative stress and chronic inflammatory response. Similar studies by others also indicate that metformin promotes healthspan and lifespan [89,90]. By decreasing LPS-induced NF-κB activity in an AMPK-dependent manner, metformin has been shown to reduce microglia and inflammatory factors [91]. Several studies indicate that metformin prevents acute lung injury in an AMPK-dependent pathway [92,93]. Most of the data from the animal models suggest that metformin prevents sepsis by significantly reducing the production of inflammatory cytokines and chemokines that contribute to inflammatory response and increase the overall lifespan.

Several studies had varying opinions on the effectiveness of metformin in septic patients. A meta-analysis study by Li et al. [94] concluded that patients who used metformin preadmission had decreased mortality from sepsis compared to patients who did not. The analysis included 8195 patients. Some studies in the meta-analysis state that metformin users had higher lactic acid and creatinine levels compared to non-metformin users; however, the meta-analysis shows no significant difference in those biomarkers. In fact, there was increasing evidence that metformin assisted the immune system by decreasing pro-inflammatory markers like NF-κB and TNF-α. This meta-analysis recognized that studies in which metformin had no significance in the pathogenesis of sepsis also had small sample sizes and various combinations of other anti-diabetic drugs. In addition to its anti-diabetic effects, metformin has been observed to inhibit inflammatory factors and induce AMPK activation, which further inhibits inflammatory immune response and improves organ recovery. This meta-analysis encourages further investigation into the use of metformin in septic patients, specifically metformin dosing.

Another cohort study includes 44 metformin users, and 118 non-metformin users admitted to the emergency room (ER) with serum lactate greater than 10 mmol/L and sepsis [95]. The metformin users had a mortality rate of 56.8% compared to an 88.1% mortality rate for the non-metformin users. This result surprised the researchers because the metformin users had considerable risk factors such as older age, diabetes, cardiovascular disease, and acute kidney injury (AKI). This study suggested that metformin changes cellular metabolism by inducing AMPK. AMPK upregulates ATP-generating pathways and downregulates ATP-consuming pathways, which have been shown to enhance cellular function during a stressed state [95].

There are some reports that suggest that metformin, when used prior to diabetic patients, has a lower risk of sepsis. A large cohort study in Korea [96] identified the association between metformin therapy and the development of sepsis in diabetic patients. This study observed 7337 patients (34,041 metformin users and 43,296 non-metformin users). Among these patients, 2512 patients (3.2%) were diagnosed with sepsis between 2011 and 2015. The analysis suggests that prior metformin therapy is not associated with the risk of sepsis and related 30-day mortality. Similarly, another meta-analytical study indicates that preadmission metformin use had decreased mortality rates in diabetic patients with sepsis [97]. They also found that compared to prior metformin use, the non-metformin users with sepsis showed no significant change in serum creatinine and lactic acid levels. Similarly, a US-based study indicated that previous metformin use decreased the 90-day mortality rate and reduced acute kidney injury in 682 sepsis patients with diabetes when compared to non-metformin users [98].

## 7. Metformin in Controlling Infectious Diseases including COVID-19

Infectious diseases caused by several microorganisms, such as bacteria, viruses, and fungi, are a significant risk of morbidity and mortality worldwide. Their prevalence is increasing daily, spreading through multiple means, including contact with other people, animals, insects, birds, and the environment. Further, new and emerging viruses have made it difficult to successfully control their invasion due to a lack of appropriate vaccines and understanding of how their elicited immune and inflammatory responses have led to widespread sickness and death. For example, the recent pandemic of COVID-19, which seems to have originated from bats, caused havoc in the world with significant hospitalizations and deaths [99]. When a foreign pathogen invades a host, the innate immune system is activated, causing widespread inflammation followed by an adaptive response. Susceptibility varies with pathogens and exposed populations, but immunocompromised patients and people with chronic conditions are particularly susceptible to these opportunistic infections. The acute inflammatory response resulting from a foreign pathogen can result in chronic inflammation. Chronic inflammation, in turn, may cause several health problems, including heart and lung diseases, arthritis, Alzheimer’s disease, and cancer. Current methods used to treat chronic inflammation due to infections include antibiotics to control the infections, anti-inflammatory agents such as corticosteroids, and the nonsteroidal anti-inflammatory drugs acetaminophen and ibuprofen to manage inflammation. Additionally, several antioxidant supplements, such as vitamins C and D, and minerals, such as zinc, have been shown to reduce inflammatory symptoms [100,101,102]. Although natural antioxidants and minerals have been shown to reduce the inflammatory response, they are not clinically approved to treat infectious diseases. Excessive use of corticosteroids has been shown to reduce the patient’s quality of life [103]. Another standard treatment to alleviate inflammation is (NSAIDs) such as aspirin or ibuprofen. However, NSAIDs have been associated with hepatic and other toxic side effects when used for a prolonged time and due to their interactions with other drugs [104]. Therefore, better, and more effective strategies are necessary to control infection-associated inflammatory responses.

Recent studies have also shown that metformin, with its potent antioxidative and anti-inflammatory actions, could prevent chronic inflammation associated with infectious diseases.

Metformin plays a direct role in mediating anti-inflammatory pathways in certain viruses and reduces the risk of complications associated with the disease. When a virus invades a host cell, pathogen recognition receptors (PPRs) recognize the pathogen-associated molecular patterns (PAMPs) displayed by the virally infected cell. This triggers the innate immune system and its inflammatory response. In HIV, metformin is believed to play a role in several signaling pathways, including suppressing NF-κB activation, which is chronically active in several inflammatory diseases, indirectly reducing a pro-inflammatory cytokine, TNF-α, and inhibiting mTOR through an AMPK-dependent manner, resulting in reduced CD4 T cell levels and decreased inflammation [105]. Further, it was found that metformin limits cytokine production from T cells. In a hepatitis-infected mouse model, metformin treatment reduced the levels of IFN-γ+ TNF-α+ and IFN-γ+IL-2+ T cells compared to the control group [106].

Metformin has also been shown to reduce ant-inflammatory pathways during bacterial infections. Abbas et al. demonstrated that metformin acts as a quorum-sensing inhibitor of PAO1, suggesting that it could be used to treat pseudomonas aeruginosa infection. The study suggests that metformin could prevent the hydrolytic enzymes produced by this pathogen and thus bacterial spread in the host body [107].

Further, some studies also suggest that metformin has anti-bacterial effects by promoting the mitochondrial ROS and inhibiting the mitochondrial complex 1, leading to the fusion of phagosomes and lysosomes [108]. Similarly, metformin has been shown to prevent Legionella pneumonia by promoting the activity of AMPK and increasing the mitochondrial ROS [109]. Metformin treatment has also been shown to improve the Mycobacterium tuberculosis infection in diabetes and non-diabetes patients [110]. Metformin has been shown to strengthen CD4+ T-cell-related chronic colitis by altering the gut microbiota as well as AMPK activation in a mouse model [111]. Further, it has been shown that metformin reduces bacterial translocation in a colitis model. Bacterial infections can cause chronic inflammation based on the type of bacteria, or they can also cause chronic inflammation if left untreated. Bacterial DNA extracted from livers, mesenteric lymph nodes (MLNs), and spleens were observed with and without metformin. The number of bacterial copies in the MLNs and liver was much lower in the metformin-treated group. Colitis and epithelial barrier dysfunction are thought to contribute to the onset and progression of many diseases, including Crohn’s and inflammatory bowel disease. It was found that metformin reduced the levels of cytokines IL-6, TNF-α, and IL-1β involved in the inflammatory pathway and inhibited the downstream AMPKα1-dependent signaling pathway in colitis models [111,112].

Further, several studies indicate that metformin helps reduce the symptoms associated with inflammatory bowel disease as well as colitis in diabetic and non-diabetic patients [113]. Moreover, type-2 diabetes patients on metformin treatment have a reduced risk of developing IBD [114]. Some studies suggest that metformin’s glucose-lowering effects are due to its interactions with the gut microbiome [115,116]. Specifically, metformin alters the gut microbiota of type-2 diabetes patients. A cross-sectional study indicates that metformin reduces butyrate-producing bacterial species such as Roseburia and Clostridiales while increasing the prevalence of Escherichia species [117]. In another study, metformin has been shown to increase mucin-degrading bacteria such as *Allermansia muciniphila* [118]. Thus, both preclinical animal studies and clinical studies clearly suggest that metformin, by altering the microbiome signatures, helps promote human health.

Recent studies have also investigated the effect of metformin in reducing the symptoms associated with COVID-19. Coronaviruses are a group of RNA viruses that are responsible for the current COVID-19 pandemic. Symptoms of COVID-19 include cough, fever, loss of taste, loss of smell, breathing difficulties, and fatigue. People with COVID-19 can develop severe, widespread inflammation leading to damaged organs and respiratory distress. One retrospective study using electronic health record data found that metformin usage is associated with decreased mortality. It was observed that metformin treatment before COVID-19 diagnosis resulted in reduced mortality for patients with diabetes and COVID-19 [119]. Similarly, a retrospective analysis study by Luo et al. [120] suggested that metformin treatment in diabetes patients decreases the mortality rate due to COVID-19. Crouse et al. [119] recommended that metformin use reduced COVID-19-associated mortality in non-diabetics and diabetics. Further, Bramante et al. [121] indicated that metformin decreased COVID-19 mortality in female subjects with obesity and diabetes. Further, a recent randomized, quadruple-blind, and placebo-controlled phase III trial suggests that outpatient treatment of metformin prevented the long COVID incidence by ~41%. Metformin also reduced COVID-19 incidence by 4.1% when compared to placebo, suggesting the beneficial effect of metformin in COVID-19 prevention [122]. Similarly, another study analyzed the possible role of metformin beyond diabetes. The study indicates that SARS-CoV-2 binds to the ACE2 receptor and metformin works through AMPK activation, leading to the phosphorylation of ACE2. It is hypothesized that adding a phosphate group would cause a conformational change in the ACE2 receptor, leading to decreased virus binding [123]. AMPK activation has also been involved in pulmonary fibrosis during severe infections, and metformin has been shown to prevent pulmonary fibrosis [124,125]. COVID-19 infection has also been shown to increase pulmonary fibrosis. Thus, metformin, by preventing pulmonary fibrosis, could alleviate symptoms of COVID-19 [126]. Further, metformin, with its antithrombotic effects and promotion of endothelial function, could also reduce the symptoms associated with COVID-19 [127]. Patients with COVID-19 have shown increased expression of various pro-inflammatory cytokines [128]. Additionally, with its potent anti-inflammatory effects, metformin prevents the expression of various cytokines, chemokines, and growth factors involved in the cytochrome storm induced by COVID-19. Few studies suggest that metformin-treated patients have low IL-6, IFN-γ, and 1L-1β levels [129,130,131]. Thus, preclinical and clinical studies indicate that metformin can potentially prevent COVID-19 symptoms when used alone or as an adjuvant with antiviral medications (Figure 5).

## 8. Conclusions

In recent years, there has been a substantial increase in studies exploring the potential uses of metformin beyond its established role in diabetes. These investigations have yielded promising results, suggesting that metformin could be repurposed to address a range of complications, including inflammation, infections, and even cancer. While the precise mechanisms through which metformin prevents these diverse complications remain to be fully elucidated, attention has been drawn to its inhibitory effect on the mitochondrial respiratory chain complex 1. The impact of metformin on AMPK activation and its interconnected relationship with cellular respiration and inflammation necessitate further exploration. A comprehensive understanding of how metformin regulates cellular energy in various human diseases holds the key to future therapeutic interventions. Notably, its beneficial role in managing symptoms associated with infectious diseases, such as COVID-19, underscores the urgency of delving deeper into the drug’s mechanisms using advanced technologies like next-generation sequencing, proteomics, and metabolomics.

To establish the efficacy of metformin, numerous epidemiological and randomized clinical studies are imperative, particularly to ascertain the benefits of both early and later administration. Population-based studies are also essential for identifying specific patient subsets that may respond more favorably to metformin treatment. Further, to gain a comprehensive insight into understanding metformin’s clinical applications besides diabetes, exploration of diverse research investigations is required. Some of these investigations must include disease-based biomarker identification, dose-dependent clinical trials, addressing the safety profile for long-term use, examining the combinational treatments, analyzing the existing real-time data obtained from the patients treated with metformin for type-2 diabetes, and exploring pharmacogenomics and pharmacodynamics. Addressing these areas could significantly contribute to a better understanding of metformin’s therapeutic implications beyond diabetes.

Although the FDA and other regulatory administrations have not approved metformin as a weight-reduction drug, several studies suggest that it can reduce weight in diabetic and prediabetic obese patients. A recent meta-analysis study indicates that treatment with metformin in different populations has a moderate effect on the reduction in BMI. Furthermore, the study also indicates that the reduction in BMI is greater in the simple obesity population when compared to before treatment [132]. Another systematic review using randomized clinical trial data indicates that metformin has modest and favorable effects on weight loss and insulin resistance with a good safety profile in children and adolescents with obesity [133]. In animal models, metformin has been shown to reduce body weight and improve metabolic activity by inhibiting white adipocyte differentiation [134]. Similarly, other studies have shown that metformin is useful in preventing weight gain by increasing the metabolic activity of brown adipose tissue [135]. However, there are no clear mechanisms that are linked with the anti-weight loss effects of metformin.

Even though metformin is primarily used for type-2 diabetes treatment, it is sometimes utilized off-label to aid weight loss or manage obesity-related conditions, particularly in individuals with insulin resistance or prediabetes. Metformin works by decreasing hepatic glucose production, increasing insulin sensitivity, and influencing appetite regulation, which could contribute to modest weight loss in some individuals. However, the magnitude of weight reduction varies among populations, and its effectiveness is increased when combined with lifestyle changes like diet and exercise. By targeting key pathways of adipose tissue dysfunction and inflammation, metformin holds promise as a therapeutic agent for mitigating obesity-related metabolic complications. However, further investigations are required to understand the mechanisms underlying metformin’s anti-inflammatory and metabolic effects in adipose tissue and to study the efficacy of metformin therapy for obesity management.

While this review primarily focuses on the drug’s implications in inflammatory and infectious diseases, it is crucial to acknowledge the wealth of information available on its role in cancer treatment. Interested readers are encouraged to please consult recent publications on metformin’s role in cancer, where evidence from epidemiological, pre-clinical, and clinical studies suggests promising implications for cancer prevention in diabetic patients [136,137,138,139]. Limited research has also explored its potential in preventing cancer in non-diabetic animal models, highlighting the need for additional studies to understand its chemopreventive effects in non-diabetics. These studies consistently highlight metformin’s significant impact on inhibiting mitochondrial respiration complex-1 and activating AMPK, influencing both cancer growth and inflammatory complications. Whether used alone or as an adjuvant, metformin holds promise as a novel therapeutic approach for preventing various inflammatory and infectious complications beyond diabetes. Combinatorial therapies, leveraging metformin’s efficacy, could enhance treatment outcomes for emerging infectious diseases. Thus, combinational therapy with metformin holds promising outcomes for treating emerging infectious diseases. This approach may benefit from existing knowledge about metformin’s antioxidative, anti-inflammatory, and immunomodulating effects, activation of AMPK, and potential antiviral properties. Adjuvant therapies could help manage comorbidities, improve treatment tolerance, and result in synergistic effects. Furthermore, combinational therapies with metformin could identify specific disease pathways and prevent secondary complications. However, rigorous clinical studies are key to validating the safety and efficacy of combinational therapies.

Besides its role in diabetes, autophagy plays a pivotal role in inflammation and infection by regulating inflammatory signaling, removing damaged organelles, and modulating immune cell function. It helps in the clearance of intracellular pathogens and regulates immune responses. Dysregulation of autophagy can lead to immune dysfunction and increased susceptibility to infections and inflammatory disorders in both diabetics and non-diabetics. Several studies suggest that metformin can modulate autophagy through multiple mechanisms [140]. Howell et al. [141] have shown that metformin activates pro-inflammatory AMPK, which inhibits the mammalian target of rapamycin complex 1 (mTORC1), a negative regulator of autophagy. By reducing mTORC1 activity, metformin promotes autophagy induction. Additionally, metformin increases lysosomal function critical for autophagosome degradation and regulates the expression of various autophagy proteins, such as Beclin-1, an important initiator of autophagy [142]. These actions collectively contribute to metformin’s modulation of autophagy. However, the precise mechanisms underlying metformin’s effects on autophagy may vary depending on the cell type and context of diabetes, necessitating further investigations to fully understand the role of metformin in regulating autophagy.

In conclusion, while numerous studies underscore metformin’s potential applications across a wide range of human diseases, the precise mechanisms of its action still need to be explored. Definitive therapeutic implications of metformin in diseases other than diabetes are warranted for further exploration through additional clinical studies. Conducting clinical studies to examine metformin’s clinical efficacy beyond diabetes involves several challenges and considerations, including the need to conduct proper clinical trials. Some of these key challenges include determining the optimal dose for metformin use in non-diabetic complications, identifying and selecting the diverse patient population with different etiologies, designing studies with specific and relevant endpoints, enduring long-term safety, identifying disease-specific biomarkers, and gaining a deeper understanding of mechanisms of action. Addressing these challenges and ensuring the applicability of such studies to non-diabetic human diseases requires a multidisciplinary approach.

## Figures and Tables

**Figure 1 ijms-25-05190-f001:**
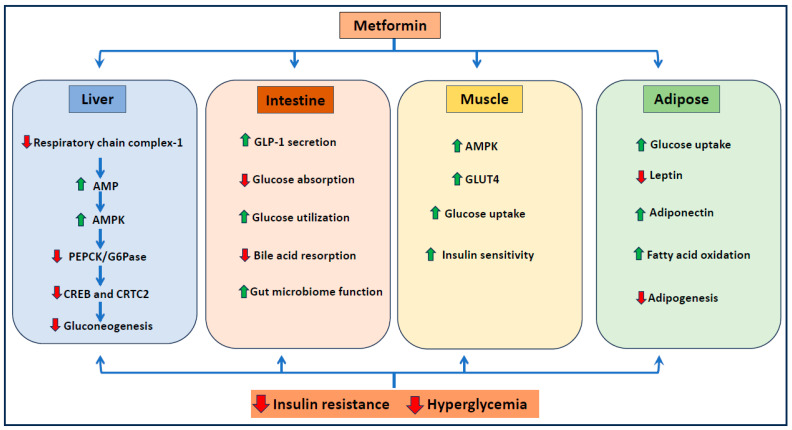
**Mechanism of action of metformin in reducing diabetes**. Metformin regulates critical components of glucose metabolism and insulin signaling pathways in various tissues. Upward arrow indicates increase and downward arrow indicates decrease. AMP: adenosine monophosphate; AMPK: AMP-activated protein kinase; CREB: cyclic AMP response element -binding protein; PEPCK: phosphoenolpyruvate carboxykinase; G6Pase: Glucose-6-phosphatase; GLUT4: glucose transporter 4; and GLP1: glucagon-like peptide.

**Figure 2 ijms-25-05190-f002:**
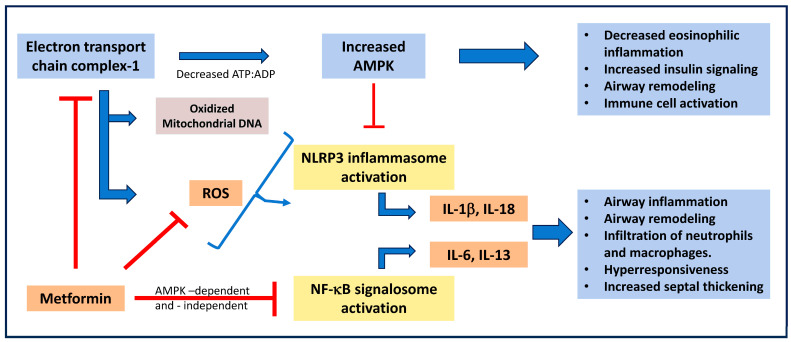
**Regulation of respiratory complications by metformin**. By preventing the mitochondrial respiratory chain complex 1, metformin increases AMPK levels. Metformin prevents activation of the NF-κB signaling pathway and NLRP3 inflammasome pathway in an AMPK-dependent and -independent manner. Thus, by preventing immune and inflammatory responses, metformin prevents airway inflammation, airway remodeling, and hyperresponsiveness during respiratory complications. AMPK: AMP-activated protein kinase; IL-1b: interleukin -1b; NF-κB: nuclear factor kappa binding protein; NLRP3: NLR family pyrin domain containing 3; and ROS: reactive oxygen species.

**Figure 3 ijms-25-05190-f003:**
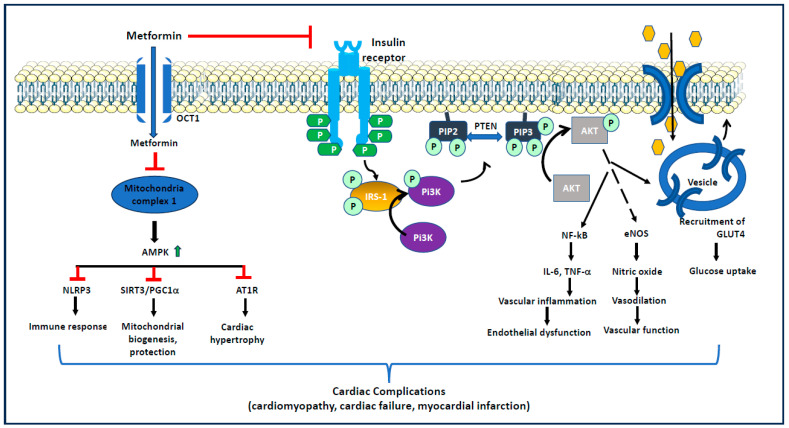
**Role of metformin in controlling cardiovascular complications**. AMPK: AMP-activated protein kinase; AKT: protein kinase B; AT1R: angiotensin 1 receptor; eNOS: endothelial nitric oxide synthase; GLUT4: glucose transporter 4; IRS1: insulin receptor substrate 1; IL-6: interleukin 6; OCT1: organic cation transporter 1; NF-kB: nuclear factor kappa binding protein; NLRP3: NLR family pyrin domain containing 3; PIP2: phosphatidylinositol 4,5-bisphosphate; PTEN: phosphatase and TENsin homolog deleted on chromosome 10; PGC1: peroxisome proliferator-activated receptor-gamma coactivator (PGC)-1alpha; PI3K: phosphatidylinositol 3-kinase; SIRT1: sirtuin 1; and TNF-α: tumor necrosis factor alpha.

**Figure 4 ijms-25-05190-f004:**
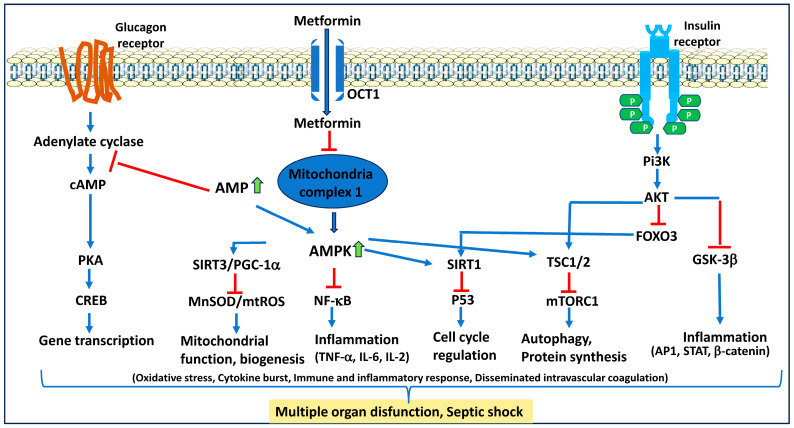
**Prevention of multiple organ dysfunction in sepsis by metformin**. Metformin prevents immune response, inflammatory response, endothelial dysfunction, disseminated intravascular coagulation, and multiple organ function in an AMPK-dependent and -independent manner. AMP: adenosine monophosphate; AMPK: AMP-activated protein kinase; cAMP: cyclin AMP; AKT: protein kinase B; CREB: cyclic AMP response element-binding protein; FOXO3: Forkhead box O3; GSK3β: glycogen synthase kinase 3 Beta; mnSOD: manganese superoxide dismutase; mtROS: mitochondrial ROS; NF-κB: nuclear factor kappa binding protein; PGC1a: peroxisome proliferator-activated receptor-gamma coactivator (PGC)-1alpha; SIRT1: sirtuin 1; TSC: tuberous sclerosis complex; and mTORC1: mammalian target of rapamycin complex 1.

**Figure 5 ijms-25-05190-f005:**
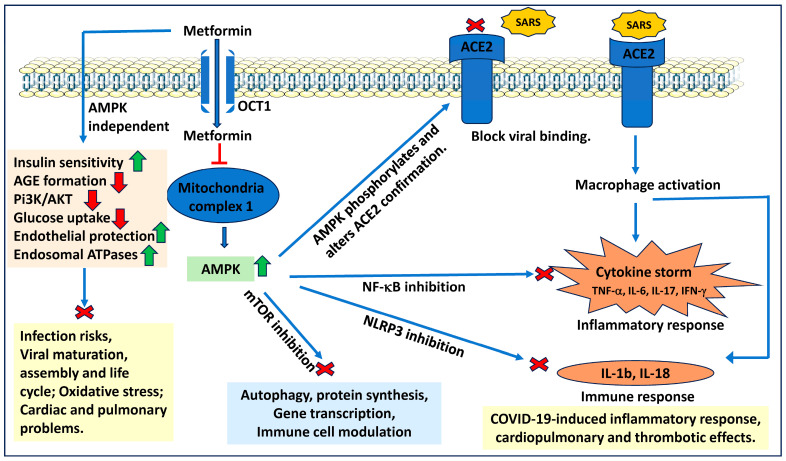
**Role of metformin in controlling COVID-19 symptoms**. Metformin has been shown to phosphorylate the ACE2 receptor and change its confirmation, leading to blockage of SARS viral binding. Metformin controls COVID-19-induced cytokine storm by preventing NF-κB-induced inflammatory response. and NLRP3-induced innate immune response in an AMPK-dependent manner. By activating the endosomal ATPases, metformin could cause viral maturation and assembly. Metformin protects from cardiopulmonary effects by increasing insulin sensitivity and endothelial function. AMPK: AMP-activated protein kinase; ACE2: angiotensin-converting enzyme 2; AKT: protein kinase B; COVID-19: coronavirus disease 2019; NF-κB: nuclear factor kappa binding protein; NLRP3: NLR family pyrin domain containing 3; PI3K: phosphatidylinositol 3-kinase; and SARS: severe acute respiratory syndrome.

## Data Availability

Not applicable.

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
