# Peer review of "Anti-Inflammatory Potential of the Anti-Diabetic Drug Metformin in the Prevention of Inflammatory Complications and Infectious Diseases Including COVID-19: A Narrative Review"

_ijms, 2024, doi:10.3390/ijms25105190_

Round 1
Reviewer 1 Report
Comments and Suggestions for Authors
The authors of review titled “Anti-inflammatory potential of anti-diabetic drug, metformin: prevention of inflammatory complications to infectious diseases including COVID-19” discussed the role of metformin, in the prevention of various inflammatory complications and some infectious diseases in both diabetics and non-diabetics”
My overall comments
The review is written well, the introduction is organized efficiently and the conclusion contains most of important knowledge that represented in the review.
Some minor corrections are required
I suggest authors need to discuss two points; the first one is about the pharmacokinetics of metformin and the second one is the relationship/effect metformin on the autophagy process
Authors need to make “et al” italic “et al” in all the manuscript.
Author Response
I suggest authors need to discuss two points; the first one is about the pharmacokinetics of metformin and the second one is the relationship/effect metformin on the autophagy process.
As suggested, we have discussed these two points in the revised manuscript (Please see highlighted sections in the introduction and conclusions)
Authors need to make “et al” italic “et al” in all the manuscript.
As suggested, we have corrected the et al in the revised manuscript.
Reviewer 2 Report
Comments and Suggestions for Authors
I read with interest the paper titled "Anti-inflammatory potential of anti-diabetic drug, metformin: prevention of inflammatory complications to infectious diseases including COVID-19." This is a well written and very complete narrative review article on the role of metformin other than diabetes.
I have few suggestions to add in the manuscript:
The first suggestion is to add the information that this is a narrative review in the title to clarification of the readers.
I didn't found any methodology for the review performed. Althrough from reading, readers could understand that this is a narrative review, Authors must state what some methodology used, where they searched, how they searched. Please provide further details on that. At least one author is refered as the responsible for the conceptualization of the article. What was the concept behind this? A small methods will improve the readiness of the paper.
Aim/Objective should be provided.
You focused very well on some roles of metformin in reducing inflamation. However, i feel that it might have other unexplored and secondary roles on that (for eg, when used in obesity, the inflamation also reduces as a secondary outcome). Some studies point out that metformin inhibits inflammation and adipocyte senescence in the obese state by regulating the adipocyte cycle program (at least in animal models). Since Metformin is used (in some countries as off-label) in the obesity, I feel that this idea could be also explored somewhere in your paper.
Author Response
The first suggestion is to add the information that this is a narrative review in the title to clarification of the readers.
As suggested, we have made changes to the title. Now the revised title is “Anti-inflammatory potential of anti-diabetic drug, metformin, in the prevention of inflammatory complications to infectious diseases including COVID-19: a narrative review.”
I didn't found any methodology for the review performed. Althrough from reading, readers could understand that this is a narrative review, Authors must state what some methodology used, where they searched, how they searched. Please provide further details on that. At least one author is refered as the responsible for the conceptualization of the article. What was the concept behind this? A small methods will improve the readiness of the paper.
As suggested, we have included methodology (search strategy) for this review. The concept is to make readers to understand the current role of metformin in regulating the inflammatory and infectious diseases.
Aim/Objective should be provided.
As suggested, we have included the objective/concept of this study in the methodology section.
You focused very well on some roles of metformin in reducing inflamation. However, i feel that it might have other unexplored and secondary roles on that (for eg, when used in obesity, the inflamation also reduces as a secondary outcome). Some studies point out that metformin inhibits inflammation and adipocyte senescence in the obese state by regulating the adipocyte cycle program (at least in animal models). Since Metformin is used (in some countries as off-label) in the obesity, I feel that this idea could be also explored somewhere in your paper.
As suggested, we have discussed briefly the role of metformin in obesity and its off-label use in the weight loss in the conclusions section. Additional citations 135-138 have been included. Please see highlighted section in the revised manuscript.